# Characterization of *Limosilactobacillus reuteri* KGC1901 Newly Isolated from *Panax ginseng* Root as a Probiotic and Its Safety Assessment

Hye-Young Yu [1], Mijin Kwon [1], Yun-Seok Lee [2], Seung-Ho Lee [1] and Sang-Kyu Kim [1,*]

[1] Laboratory of Efficacy Research, Korea Ginseng Corporation, 30 Gajeong-ro, Yuseong-gu, Daejeon 34128, Republic of Korea
[2] Laboratory of Products, Korea Ginseng Corporation, 30 Gajeong-ro, Yuseong-gu, Daejeon 34128, Republic of Korea
* Correspondence: 20100366@kgc.co.kr; Tel.: +82-10-9520-4492

**Abstract:** In this study, we determined the probiotic properties and safety of *Limosilactobacillus reuteri* KGC1901 isolated from Korean *Panax ginseng* root. This strain was identified based on its 16s rRNA sequence, and the absence of genes related to antibiotic resistance and virulence was confirmed through whole genome analysis in terms of safety. Moreover, this strain had no antibiotic resistance to eight antibiotics as proposed by the European Food Safety Authority, did not show hemolytic activity on blood agar, and did not produce biogenic amines. *L. reuteri* KGC1901 also showed the capability to survive at low pH and in presence of bile salts and sufficiently adhered to HT-29 cells and mucin. The adhesion ability to HT-29 was confirmed by immunofluorescence staining. These results indicated the excellent viability of *L. reuteri* KGC1901 in the human gastrointestinal tract. Additionally, *L. reuteri* KGC1901 had antimicrobial activity against *Clostridium difficile*, and the presence of the reuterin (3-HPA) biosynthetic gene cluster in the genome was revealed. Furthermore, the cell-free supernatant of KGC1901 had DPPH scavenging activity and reduced the nitric oxide production in LPS-stimulated RAW 264.7 cells. Based on these results, it was confirmed that *L. reuteri* KGC1901 derived from ginseng has sufficient potential to be used as a probiotic.

**Keywords:** probiotics; lactic acid bacteria; *Limosilactobacillus reuteri*; *Panax ginseng*; reuterin; antioxidant





## 1. Introduction

Microorganisms comprising the human microbiome, including bacteria, archaea, viruses, and eukaryotes, are ubiquitous in the body, including the skin, gastrointestinal tract, respiratory tract, and mammary gland, and interact with their living host [1]. Through this symbiosis, these microbes are closely related to health and disease in humans. The World Health Organization defines probiotics as "live microorganisms which when administered in adequate amounts confer a health benefit on the host" [2]. The most common and traditional genera of probiotic microorganisms are *Bifidobacterium* and strains belong to lactic acid bacteria (LAB) such as *Lactobacillus*, *Lactococcus*, *Pediococcus*, *Enterococcus*, and *Streptococcus* [3,4]. Recent studies also have reported *Prevotella*, *Akkermansia*, *Feacalibacterium*, and *Bacteroides* as next-generation probiotics [5]. These probiotics play several important roles in the human body such as immune boosting and enhancing metabolic syndromes by regulating the intestinal environment [5–8]. Previous studies have also shown that probiotics induce the maturation of intestinal epithelium [9]. Nonetheless, as a commercial product, safety remains a major issue. Therefore, probiotic microorganisms should also be evaluated for safety characteristics such as hemolytic activity, antibiotic resistance, and production of toxic metabolites [10,11]. Traditionally, most probiotic strains including *Limosilactobacillus reuteri* have been isolated from the human gut because these microbes should be colonized and survive in the human gastrointestinal tract [12,13]. However,

recently, it has been discovered that autochthonous LAB from raw fruits and vegetables also have several functional features as probiotics: resistance to gastric juice and bile acid, adhesion ability to gut epithelium, and ability to modulate the human immune system. Beatrice et al. reported probiotic potentials of 48 strains of lactic acid bacteria isolated from French beans, carrots, tomatoes, pineapple, cauliflower, and celery [12]. Arshad et al. also reviewed the probiotic properties and human health beneficial effects of *Lactobacilli* originating from various sources including plant and vegetable matrices [14].

Korean ginseng (*Panax ginseng* C.A. Meyer) is a well-known traditional herbal medicine that has been used for centuries in East Asia. Ginsenosides, the main active ingredients of ginseng, are known to be effective in oxidative stress and cardiovascular dysfunction [15]. Bacterial endophytes, which have been found in every part of the ginseng plant, including the roots, stem, leaves, and fruits, are known to have the potential to produce ginsenosides and the ability to transform major ginsenosides into a minor form using enzyme activities [16]. The abundant phyla of endophytic bacteria from Korean ginseng are *Proteobacteria*, *Actinobacteria*, and *Firmicutes*, and the dominant endophytic bacterial genera are *Bacillus* and *Pseudomonas* [16]. Raw ginseng has a highly acidic matrix and is rich in dietary fiber, so it is suitable for the survival and proliferation of endophytic LAB [17,18]. In recent studies, *Lacticaseibacillus paracasei*, *Limosilactobacillus fermentum*, and *Lacticaseibacillus casei* were isolated from *P. ginseng* roots in Korea and the health beneficial effects, as well as the probiotic properties and safety of these probiotic bacteria, have been reported [6,19,20].

*Clostridium difficile* is a Gram-positive, anaerobic, spore-forming bacterium that infects the intestinal tract and causes severe diarrhea and potentially life-threatening complications resulting in death [21]. Antibiotics such as vancomycin or fidaxomicin are administered primarily for the treatment of *C. difficile* infection, but these broad-spectrum antibiotics eventually destroy the balance of the intestinal microflora and cannot be a fundamental treatment method [22,23]. Nowadays, the application of probiotics has emerged as an alternative to antibiotic therapy [24].

Reactive oxygen species (ROS) are formed naturally as a byproduct of the normal metabolism of oxygen and play important roles in cell signaling [25,26]. However, the overproduction of ROS may cause oxidative stress resulting in cell damage. Eventually, oxidative stress can cause a wide range of diseases such as cancer, aging, and Alzheimer's disease [27,28]. Supplementation of exogenous antioxidants may sometimes be required to reduce such oxidative stress. In the gut environment, lactic acid bacteria play an important antioxidant role by maintaining the intestinal redox balance [29]. Averina et al. studied the antioxidant potential of *lactobacillus* and *Bifidobacterium* and their genetic and biochemical mechanisms [30].

In this study, we isolated *Limosilactobacillus reuteri* KGC1901 from ginseng root, investigated its probiotic properties, and assessed its safety as a commercial probiotic. We also explored its antimicrobial activity against *Clostridium difficile* and its antioxidant activity.

## 2. Materials and Methods

### 2.1. Isolation of LAB Strains from Raw Ginseng and Identification of L. reuteri KGC1901

We isolated endophytic LAB from 6-year-old ginseng roots (*Panax ginseng* C.A. Meyer) collected from Anseong, Korea. The collected roots were washed to remove soil debris and then sterilized in 70% ethanol and 3% NaClO. The root samples were sliced and then mixed with a sugar solution containing 1.25% ($w/v$) sucrose and 2.5% ($w/v$) glucose. The mixture was fermented at 37 °C for 5 days; then, the fermented solution was serially diluted in 9 mL sterile saline, and the diluted samples were spread on MRS agar medium (BD Difco, Franklin Lakes, NJ, USA) containing 0.5% ($w/v$) $CaCO_3$. After incubation at 37 °C for 24–48 h, we selected candidate colonies with a halo around the colony. Then, the colonies were identified based on their 16s rRNA sequence (Solgent, Daejeon, Republic of Korea) and deposited in the Korea Collection for Type Culture (KCTC, Jungeup, Republic of Korea). The strain was stored in 25% ($v/v$) glycerol at −80 °C. The whole genome of *L. reuteri* KGC1901 was sequenced on the Illumina MiSeq 300 system with 2 × 300 bp paired-end

reads using the 600-cycle sequencing kit (MiSeq Reagent Kit v3, CJ Bioscience, Inc., Seoul, Republic of Korea). The genome analysis was performed using the EzBioCloud database (CJ Bioscience, Inc., Seoul, Republic of Korea).

### 2.2. Carbohydrate Utilization and Enzymatic Activities

The carbohydrate utilization patterns of the strains were determined using the API 50 CHL/CH kit (bioMérieux, Marcy-l'Etoile, France) according to the manufacturer's instructions. The cell pellets resuspended in 10 mL of API 50 CHL medium were loaded into the API 50CH test strip. The utilization patterns for 49 types of carbohydrates were determined according to the manufacturer's color reaction chart after 48 h at 37 °C. Enzymatic activities were assessed using the API ZYM kit (bioMérieux, Marcy-l'Etoile, France) according to the manufacturer's instructions. The media of KGC1901 cultured for 24 h was loaded into the API ZYM strip. After 3 h at 37 °C, a drop of ZYM-A and ZYM-B reagents was added to each well, the mixture was incubated for 5 min at 25 °C, and then the resulting color changes were observed. Enzymatic activities were determined according to the manufacturer's color reaction chart. In the above two experiments, the type strain, *Limosilactobacillus reuteri* KCTC3594 purchased from KCTC, was used as a control.

### 2.3. Acid and Bile Salt Resistance

Acid and bile salt resistance were determined as described by Kim et al. [6], and *Lacticaseibacillus rhamnosus* GG (LGG) KCTC5033 purchased from KCTC served as a control. Bacterial cell pellets were collected and then resuspended in 10 mM PBS adjusted pH 2.5 and 10 mM PBS with 0.1% ($w/v$) oxgall, respectively. After incubation at 37 °C for 3 h, the numbers of viable colonies were counted, and the survival rates were determined as the following:

$$\text{Survival rate (\%)} = C_{survive}/C_{initial}$$

where $C_{survive}$: colony counts after 3 h and $C_{initial}$: colony counts before exposure to acid or oxgall.

### 2.4. Intestinal Adhesion and Mucin Binding Ability

The intestinal adhesion property using HT-29 cells (human colonic epithelial cell) was evaluated as described by Kim, et al. [6]. The cultured HT-29 cells ($4 \times 10^5$ cells/mL) were incubated in RPMI 1640 medium (Hyclone, Logan, UT, USA) without fetal bovine serum and antibiotics at 37 °C in a 5% $CO_2$ air overnight in 12-well tissue culture plates (SPL, Pocheon, Republic of Korea). Each probiotic strain ($1 \times 10^7$ cells/mL) was added to the HT-29 cell monolayer and incubated at 37 °C in 5% $CO_2$ air. After 2 h of incubation, unattached bacterial cells were removed by PBS, and attached bacterial cells were detached by 0.05% trypsin-EDTA solution ($w/v$). Mucin binding ability was determined as described by Yasmin et al. with slight modification [31]. In brief, 0.4% mucin solution from porcine stomach type II (Sigma-Aldrich, St Louis, MO, USA) was coated to a 12-well non-coated plate for 24 h at 4 °C. Each probiotic strain ($1 \times 10^7$ cells/mL) was added to a mucin-coated well and incubated for 2 h at 37 °C in 5% $CO_2$ air. Next, unattached bacterial cells were removed and washed with PBS, and attached bacterial cells were detached by 0.05% trypsin-EDTA solution ($w/v$). The number of detached viable bacterial cells was measured as follows:

$$\text{Adhesion (\%)} = A_1/A_0 \times 100$$

where $A_1$: number of attached bacterial cells and $A_0$: number of initial bacterial cells.

In the above two experiments, *Limosilactobacillus reuteri* KCTC3594, the type strain purchased from KCTC, was used as a control. All data were obtained from three independent experiments and expressed as mean ± standard error. Significant differences were determined using Student's *t*-tests and indicated as * $p < 0.05$ and ** $p < 0.01$.

### 2.5. Immunofluorescence Staining

HT-29 cells were cultured at a density of $2 \times 10^5$ cells/well in a 4-well cell culture slide (SPL, Republic of Korea) for 1 day. Lactic acid bacteria were diluted to $1 \times 10^6$ CFU/mL with RPMI and inoculated into the HT-29 cells for 2 h. The cells were washed with PBS then fixed by 4% paraformaldehyde and permeabilized with 0.1% Triton X-100 (Sigma-Aldrich, Germany), and they were blocked in 1% Bovine Serum Albumin (Sigma-Aldrich, Germany) for 1 h. Then, they were treated with peptidoglycan antibody (Bio-Rad, Hercules, CA, USA) overnight at 4 °C and goat anti-mouse IgG H&L Alexa Fluor 488 as a secondary antibody for 1 h at room temperature. The slide was mounted using a mounting medium with DAPI (Abcam, Cambridge, UK) and visualized with a confocal laser scanning microscope (LSM 800, Carl Zeiss, Jena, Germany).

### 2.6. Antibiotic Resistance and Virulence Factor

The European Food Safety Authority (EFSA) has published cut-off MIC values for the following antibiotics: ampicillin, gentamicin, kanamycin, streptomycin, erythromycin, clindamycin, tetracycline, and chloramphenicol. We determined the corresponding MIC values for KGC1901 by performing E-tests [32]. In brief, KGC1901 was spread onto an MRS agar plate, and a test strip (Liofilchem, Roseto degli Abruzzi, Italy) was applied aseptically to the inoculated surface of the agar plate. Following incubation at 37 °C for 24 h, the inhibition zone formed around the strip was observed, and the MIC values were determined on the basis of the concentration of the antibiotic on the strip where the edges of the inhibition zone converged. Acquired antimicrobial resistance genes were detected by comparing the assembled sequences with the reference sequences using ResFinder v.4.1 (https://cge.cbs.dtu.dk/servies/ResFinder (accessed on 1 December 2022)) with a threshold of >90% for %ID and 60% for minimum length [33]. Putative virulence genes were identified using the Virulence Factor Database (version 2020.02.13; http://www.mgc.ac.cn/VFs/ (accessed on 1 December 2022)) by the BLASTn algorithm with conditions of identity > 70%, coverage > 70%, and E-value < $1 \times 10^{-5}$ [34].

### 2.7. Determination of Hemolytic Activity and Biogenic Amine

For determination of hemolytic activity, *Escherichia coli* KCTC2441 purchased from KCTC and *Staphylococcus aureus* NCTC10788 from bioMérieux (Marcy-l'Etoile, France) were used as positive controls of α-hemolysis and β-hemolysis, respectively. Each strain was streaked on blood agar plates (KisanBio, Seoul, Republic of Korea), then hemolysis types were determined after incubation. The production of biogenic amines (BAs) was measured as described by Bang et al. [35]. Briefly, BAs from probiotic cell-free supernatant (CFS) were derivatized and analyzed on a high-performance liquid chromatography system (LC-NETI/ADC, Jasco, Macclesfield, UK) equipped with a C18 column (ANPEL Laboratory analysis, Shanghai, China, 4.6 mm × 250 mm C18 column) with a detection wavelength of 254 nm. The mobile phase consisted of an aqueous acetonitrile solution (67:33 of water) flowing at a rate of 0.8 mL/min.

### 2.8. Antimicrobial Activity of L. reuteri KGC1901 CFS against Clostridium difficile

*Clostridium difficile* type strain was purchased from the National Culture Collection for Pathogens (Korea) and was cultured in BHI medium (BD Difco, Franklin Lakes, NJ, USA) for 18 h at 37 °C under anaerobic conditions. Then, 1% (*v/v*) of the cultured broth of *C. difficile* was transferred to fresh BHI medium with 10%, 20%, and 40% (*v/v*) of KGC1901 CFS, and the suspensions were incubated for 48 h at 37 °C under anaerobic conditions. Antimicrobial activity was determined by measuring the differences in the absorbance of cultures at 600 nm at the start and end of the incubation period with or without CFS.

### 2.9. DPPH Radical Scavenging Activity

The antioxidant activity of KGC1901 was determined on the basis of DPPH radical scavenging activity. An overnight culture of KGC1901 was centrifuged at 10,000 rpm for

10 min at 4 °C. The cells were collected, washed twice with PBS, and resuspended in PBS with an absorbance of 1.0 at 600 nm. The cell suspension and DPPH solution (200 μM) were mixed and incubated at 37 °C for 30 min. The mixture was centrifuged at 10,000 rpm for 5 min, and the absorbance of the supernatant was measured at 517 nm. Ascorbic acid was used as the positive control, and scavenging activity was expressed as the percentage of inhibition of the sample relative to the control.

### 2.10. Cell Culture and Nitric Oxide (NO) Production Assay

The mouse macrophage cell line RAW 264.7 was obtained from the Korean Cell Line Bank (KCLB40071), and the cells were cultured in Dulbecco's modified Eagle's medium (Gibco, Waltham, MA, USA) supplemented with 10% ($v/v$) FBS (Gibco) and 1% ($w/v$) penicillin/streptomycin solution (Gibco). The cells were incubated at 37 °C under a humidified atmosphere of 5% $CO_2$. RAW 264.7 cells ($5.0 \times 10^5$ cells/mL) were exposed to 1 μg/mL of lipopolysaccharide (LPS), and then aliquots were treated with 1% of cell-free supernatant obtained from *L. reuteri* KGC1901 and *L. reuteri* KCTC14299BP, a positive control purchased from KCTC. NO production was measured using a Griess reagent system as described [36].

## 3. Results and Discussion

### 3.1. Isolation and Identification of L. reuteri KGC1901 from Ginseng Root

Based on previous studies that showed endogenous bacteria exist in all parts of ginseng plants [16], we tried to isolate endogenous lactic acid bacteria from ginseng roots. To isolate endophytic LAB from ginseng root, 6-year-old ginseng root samples were fermented and 16s rRNA sequences of candidate colonies forming halos in MRS agar supplemented with $CaCO_3$ were analyzed. Among the candidates, *Limosilactobacillus reuteri*, which is one of the valuable commercial probiotic strains, was identified, named *L. reuteri* KGC1901, and the strain was deposited in the Korea Collection for Type Culture (KCTC, Jungeup, Republic of Korea) under the accession number KCTC14652BP (Figure 1).

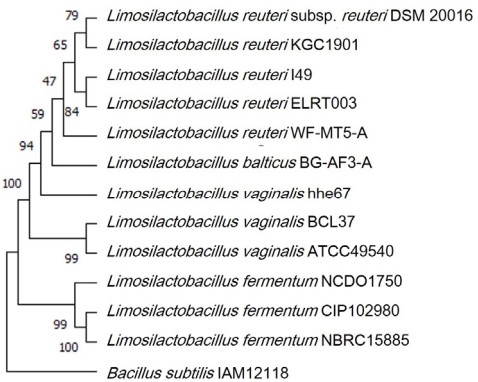

**Figure 1.** The phylogenetic tree is based on the 16s rRNA gene sequence data obtained from the NCBI database, showing the phylogenetic positions of *L. reuteri* KGC1901. The tree was constructed using the maximum likelihood algorithm and Kimura 2-parameter model and the numbers displayed next to the branch indicate bootstrap values (500 replicates). *Bacillus subtilis* was used as an outgroup. Phylogenetic analysis was conducted in MEGA11 (Molecular Evolutionary Genetics Analysis) [37].

To determine the genetic characteristics of this strain, whole genome sequencing was performed. The general features and a map of its circular genome are shown in Table 1 and Figure 2, respectively. The length of the genome is approximately 1.96 million base pairs (38.7% G + C content), containing 1944 coding DNA sequences (CDS), 10 rRNA genes, and 61 tRNA genes.

**Table 1.** Genomic characteristics of *Limosilactobacillus reuteri* KGC1901.

| Items | Results |
|---|---|
| Genome size (bp) | 1,963,544 |
| GC contents (%) | 38.7 |
| CDS | 1944 |
| No. of rRNA genes | 10 |
| No. of tRNA genes | 61 |

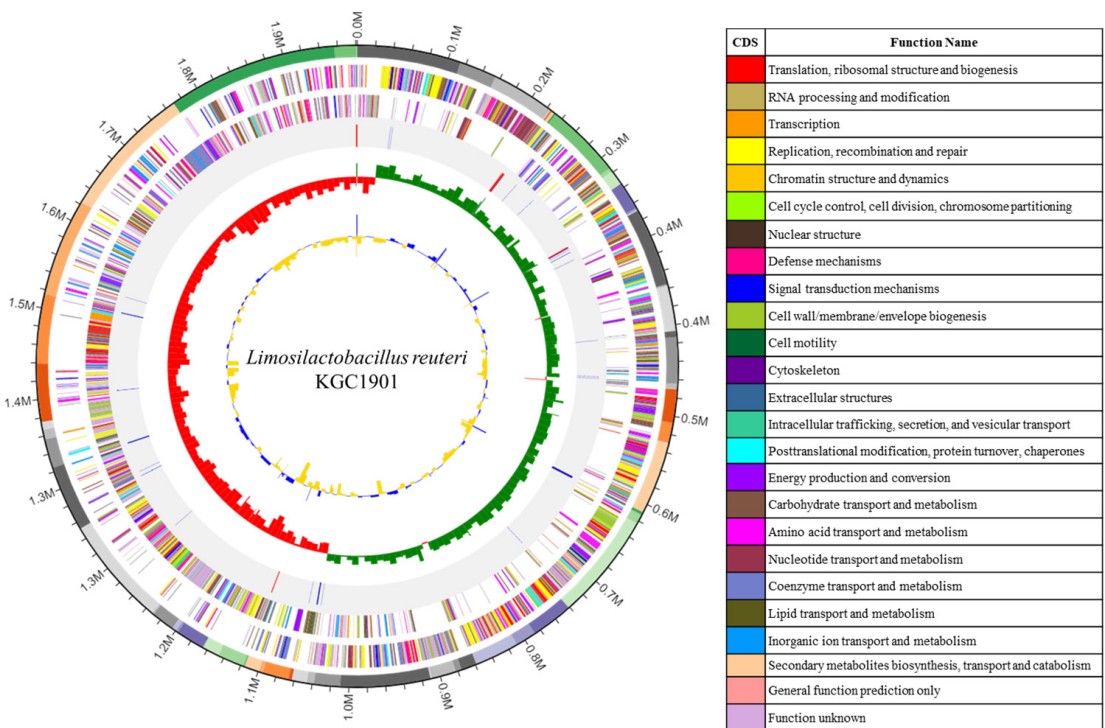

**Figure 2.** Genome map of *L. reuteri* KGC1901. From outside to the center: 71 contigs, annotated reference genes (specifically, CDSs) found in the forward strand; in the reverse strand, rRNA and tRNA found in this genome; GC skew metric (higher-than-average values: green, lower-than-average values: red); and GC ratio metric (higher-than-average values: blue, lower-than-average values: yellow).

### 3.2. Carbohydrate Fermentation and Enzymatic Activities

Among 49 carbohydrates, KGC1901 and KCTC3594 differed in their fermentation patterns only on D-mannose; specifically, KGC1901 fermented D-mannose, whereas KCTC3594 did not (Table 2). These results can be used by the industry to optimize culture conditions for enhancing productivity. Microbial enzymes, such as β-glucosidase and β-glucuronidase, can generate carcinogens or increase the possibility of carcinogenesis in the human body [38,39]. Hence, evaluating the activities of extracellular microbial enzymes is critical for product safety. Our results show the absence of β-glucosidase and β-glucuronidase activities in KGC1901 (Table 3). The activities of enzymes involved in carbohydrate metabolism (α-galactosidase, β-galactosidase, and α-glucosidase), lipid metabolism (esterase), and vitamin metabolism (acid phosphatase) were detected in KGC1901. In particular, β-galactosidase and leucine arylamidase activities were stronger than other enzyme activities.

**Table 2.** Carbohydrate fermentation of *L. reuteri* KGC1901.

| Carbohydrates | Strains | | Carbohydrates | Strains | |
|---|---|---|---|---|---|
| | KGC 1901 | KCTC 3594 | | KGC 1901 | KCTC 3594 |
| Glycerol | - * | - | Salicin | - | - |
| Erythritol | - | - | D-cellobiose | - | - |
| D-arabinose | - | - | D-maltose | + | + |
| L-arabinose | + | + | D-lactose | + | + |
| D-ribose | + | + | D-melibiose | + | + |
| D-xylose | - | - | D-saccharose(sucrose) | + | + |
| L-xylose | - | - | D-trehalose | - | - |
| D-adonitol | - | - | Inulin | - | - |
| Methyl-βD-Xylopyranoside | - | - | D-melezitose | - | - |
| D-galactose | + | + | D-raffinose | + | + |
| D-glucose | + | + | Amidon (starch) | - | - |
| D-fructose | - | - | Glycogen | - | - |
| D-mannose | + | - | Xylitol | - | - |
| L-sorbose | - | - | Gentiobiose | - | - |
| L-rhamnose | - | - | D-turanose | - | - |
| Dulcitol | - | - | D-lyxose | - | - |
| Inositol | - | - | D-tagatose | - | - |
| D-mannitol | - | - | D-fucose | - | - |
| D-sorbitol | - | - | L-fucose | - | - |
| MethylαD-mannopyranoside | - | - | D-arabitol | - | - |
| MethylαD-Glucopyranoside | - | - | L-arabitol | - | - |
| N-acetylglucosamine | - | - | Potassium gluconate | v | v |
| Amygdalin | - | - | Potassium 2-ketogluconate | - | - |
| Arbutin | - | - | Potassium 5-ketogluconate | - | - |
| Esculin ferric citrate | + | + | | | |

* -: not utilized; v: weakly utilized; +: strongly utilized.

**Table 3.** Enzymatic activities of *L. reuteri* KGC1901.

| Enzymes | KGC1901 | Enzymes | KGC1901 |
|---|---|---|---|
| Alkaline phosphatase | - * | Naphthol-AS-BI-phosphohydrolase | + |
| Esterase | + | α-Galactosidase | + |
| Esterase lipase | - | β-Galactosidase | +++ |
| Lipase | - | β-Glucuronidase | - |
| Leucine arylamidase | +++ | α-Glucosidase | + |
| Valine arylamidase | + | β-Glucosidase | - |
| Crystinearylamidase | - | N-acetyl-β-glucosaminidase | - |
| Trypsin | - | α-Mannosidase | - |
| α-Chymotrypsin | - | α-Fucosidase | - |
| Acid phosphatase | + | | |

* -, no activity; +, poor activity; ++, moderate activity; +++, strong activity.

### 3.3. Probiotic Properties

The gastrointestinal tract of humans is a highly stressful environment for probiotics in terms of oxygen level, nutrient limitations, and pH changes (particularly high acidity) [40,41]. The ability to tolerate acids and bile salts is essential for probiotics to survive in intestinal harsh environments and to ensure their functionalities. In previous studies, only 6 out of 13 probiotic strains isolated from wheat-bran sourdough survived exposure to low pH and bile salts [42]. *L. reuteri* KGC1901 showed 81.97% and 92.04% survival rates against exposure to pH 2.5 and oxgall, respectively (Table 4). The acid resistance of *L. reuteri* KGC1901 was similar to that of LGG (84.19%) ($p = 0.366$), which is one of the most popular probiotic strains in the industrial field and used as a positive control in this

study, and its bile salt tolerance is somewhat higher than that of LGG (88.05%) ($p = 0.066$), under the same conditions.

**Table 4.** Acid resistance, bile salt tolerance, and antioxidant activity of *L. reuteri* KGC1901. Data are presented as mean $\pm$ standard deviation values from triplicate experiments.

| Characteristics | | Survival Rate (%) | |
|---|---|---|---|
| | | **KGC1901** | **LGG** |
| Acid resistance | 0 h (log CFU/mL) | 8.26 $\pm$ 0.00 | 8.03 $\pm$ 0.06 |
| | 3 h (log CFU/mL) | 6.78 $\pm$ 0.11 | 6.77 $\pm$ 0.13 |
| | Survival rate (%) | 81.97 $\pm$ 1.38 | 84.19 $\pm$ 2.33 |
| Bile salt tolerance | 0 h (log CFU/mL) | 8.26 $\pm$ 0.00 | 8.03 $\pm$ 0.06 |
| | 3 h (log CFU/mL) | 7.61 $\pm$ 0.10 | 7.07 $\pm$ 0.02 |
| | Survival rate (%) | 92.04 $\pm$ 1.19 | 88.05 $\pm$ 0.96 |
| DPPH radical scavenging (%) | | 13 | 9 |

After passing through the gastrointestinal tract, the probiotics should be attached to the intestinal mucosa to colonize and proliferate [40]. This intestinal adhesion ability can be assessed mainly based on adhesion to HT-29 or Caco-2, human epithelial cell lines, and attachment to the intestinal mucus directly, in vitro [40]. In this study, we used HT-29 cells and 0.4% mucin solution from porcine stomach type II. After 2 h of adhesion, 51% of *L. reuteri* KGC1901 cells were attached to mucin and 19% of cells were attached to HT-29, whereas 13% and 7% of the cells of *L. reuteri* KCTC3594 were attached to mucin and HT-29, respectively (Figure 3A). The immunofluorescent image also confirmed that more KGC1901 cells adhered to HT-29 than the type strain (Figure 3B). This indicates that *L. reuteri* KGC1901 had better adhesion ability to the intestine than the type strain. These results suggest that *L. reuteri* KGC1901 has the ability to survive in the human gut and has proper properties as a probiotic.

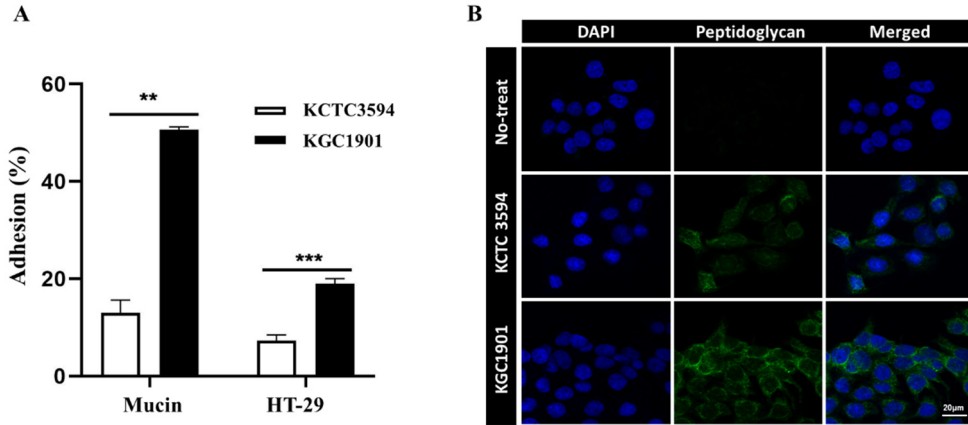

**Figure 3.** Adhesion ability and immunofluorescent image of *L. reuteri* KGC1901. (**A**) Adhesion ability of *L. reuteri* KGC1901 and *L. reuteri* KCTC3594 (positive control) to mucin and HT-29 cells. Data are presented as mean $\pm$ standard deviation values from triplicate experiments. Significant correlations between the two strains are shown by asterisks (** $p < 0.01$, and *** $p < 0.001$). (**B**) HT-29 cells were stained with DAPI (blue), and the bacterial cells of *L. reuteri* KGC1901 and *L. reuteri* KCTC3594 were stained with antibodies against peptidoglycan and goat anti-mouse Alexa fluor 488 (green).

### 3.4. Safety Assessments

Whole-genome sequencing is as important as biochemical tests in that it can predict the presence of risk factors [43]. In this study, whole-genome sequencing was performed to understand the safety of *L. reuteri* KGC1901 genetically, and the whole-genome sequences were analyzed using ResFinder and VFDB to identify antibiotic resistance genes and virulence genes, respectively. As a result, the genome of *L. reuteri* KGC1901 had no putative

genes related to antibiotic resistance and virulence. The horizontal transfer of antibiotic resistance genes from commercial probiotics to pathogenic bacteria via plasmid-like mobile elements can be devastating to human health [43,44]. Therefore, the fact that *L. reuteri* KGC1901 strain does not have an antibiotic resistance gene is very valuable for a safe probiotic material. The antibiotic resistance of *L. reuteri* KGC1901 was confirmed by measuring the MICs of eight antibiotics together with confirmation at the genome level. As a result, *L. reuteri* KGC1901 showed MICs below the cut-off values suggested by EFSA for eight antibiotics: ampicillin, gentamicin, kanamycin, streptomycin, erythromycin, clindamycin, tetracycline, and chloramphenicol. Finally, we determined that it is safe against antibiotic resistance (Table 5).

**Table 5.** Minimal inhibitory concentrations of *L. reuteri* KGC1901.

| Antibiotic [1] | AMP | GEN | KAN | STR | ERY | CLI | TET | CHL |
|---|---|---|---|---|---|---|---|---|
| Cut-off value (mg/L) [2] | 2 | 8 | 64 | 64 | 1 | 1 | 16 | 4 |
| Observed MICs | 0.016 | 3 | 48 | 8 | 0.125 | <0.016 | 1.0–1.5 | 0.75 |
| Assessment | S [3] | S | S | S | S | S | S | S |

[1] AMP, ampicillin; GEN, gentamycin; KAN, kanamycin; STR, streptomycin; ERY, erythromycin; CLI, clindamycin; TET, tetracycline; CHL, chloramphenicol; [2] European Food Safety Authority Guidelines, 2018; [3] S, Susceptible.

Since hemolytic activity is related to the virulence of bacteria, hemolysis is an important factor that must be considered when selecting a probiotic strain [31]. KGC1901 showed γ-hemolysis (i.e., no hemolytic activity on sheep blood agar), whereas *E. coli* and *S. aureus* showed α- and β-hemolysis, respectively (Figure 4).

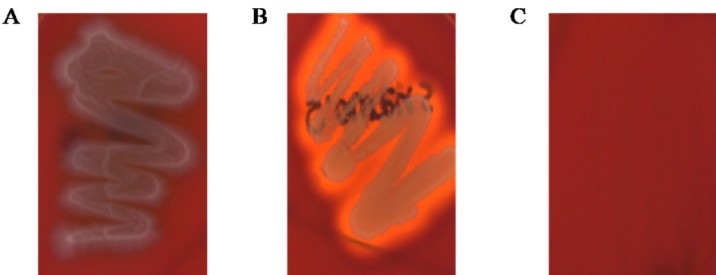

**Figure 4.** Hemolytic activities of *L. reuteri* KGC1901, *Escherichia coli*, and *Staphylococcus aureus*. *E. coli* shows typical α-hemolysis (**A**), and *S. aureus* shows typical β-hemolysis (**B**). However, *L. reuteri* KGC1901 (**C**) does not show any lysis of blood cells.

Biogenic amines (BAs), the biogenic substances containing one or more amine groups, are produced by some lactic acid bacteria during fermentation [45–47], and they normally function as neurotransmitters and signal molecules between commensal microbiota in the host gut [48]. However, BAs are also precursors for carcinogenic N-nitroso compounds, and high levels of BAs may cause symptoms such as headache, heart palpitations, vomiting, diarrhea, and hypertensive crises in humans and animals [49]. In this study, BAs such as tyramine, histamine, putrescine, cadaverine, and 2-phenethylamine were not detected in the supernatant of KGC1901. Based on these results, it was confirmed that KGC1901 has sufficient safety as a probiotic strain.

### 3.5. Antimicrobial Activity against Clostridium difficile

As previously described, the disruption of gut microbiome diversity by broad-spectrum antibiotics is a major risk factor for infection with CD in the gut [50,51]. Reuterin, also known as 3-hydroxypropionaldehyde (3-HPA), a secondary metabolite of *L. reuteri*, induces ROS production in CD, and this oxidative stress leads to cell membrane disruption, DNA damage, and finally cell death of this pathogen [52,53]. Based on these findings, in this study, the antibacterial activity of *L. reuteri* KGC1901 isolated from ginseng against CD was tested. As a result, when 10% or more of KGC1901 CFS was added to CD culture

media, the absorbance of culture broth was significantly decreased after 48 h compared to the initial time (Figure 5). Reuterin is produced by the bioconversion of glycerol by vitamin B12-dependent glycerol dehydratase [54–56]. The biosynthesis of reuterin is controlled by a gene cluster, also known as the *pdu-cbi-cob-hem* gene cluster, consisting of the glycerol/propanediol utilization (pdu) cluster and the cobalamin biosynthesis cluster. Additionally, glycerol dehydratase is encoded in *pduCDE* [57]. In this study, the presence of the *pdu* operon including *pduCDE* in contig 69 of the KGC1901 genome was confirmed through whole-genome analysis. Moreover, *cbi*, *cob*, and *hem* genes were also detected in contig 68 (Figure 6). Greppi et al. studied the presence and composition of the *pdu-cbi-cob-hem* gene cluster in the genomes of 65 *L. reuteri* strains [58]. Most strains possessing complete gene clusters produced 3-HPA, but some strains failed to produce 3-HPA even though they had complete gene clusters. In another study, Yang et al. reported on strain *L. reuteri* AN417, which does not synthesize reuterin because it lacks a gene cluster encoding reuterin in its genome but has antibacterial activity against oral pathogens. It was confirmed that the antibacterial activity of AN417 was caused by fatty acids or sugars instead of reuterin [59]. Although the mechanism(s) involved in the antimicrobial activity of KGC1901 against *C. difficile* were not elucidated in this study, the change in absorbance of CD culture media and the presence of reuterin biosynthetic genes in the genome revealed the antimicrobial activity of KGC1901.

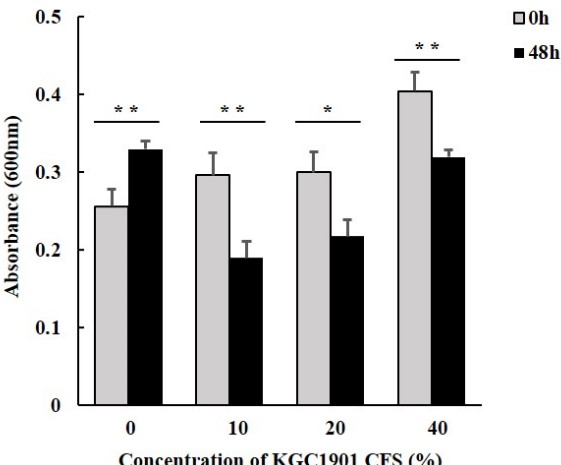

**Figure 5.** Antimicrobial activity of *L. reuteri* KGC1901 against *Clostridium difficile*. *C. difficile* was cultured in BHI with KGC1901 CFS and the absorbance of the culture soup was measured. As a result, the growth of CD was suppressed on the media added with 10%, 20%, and 40% of KGC1901 CFS. Significant correlations between 0 h and 48 h of each concentration are shown by asterisks (* $p < 0.05$, and ** $p < 0.01$).

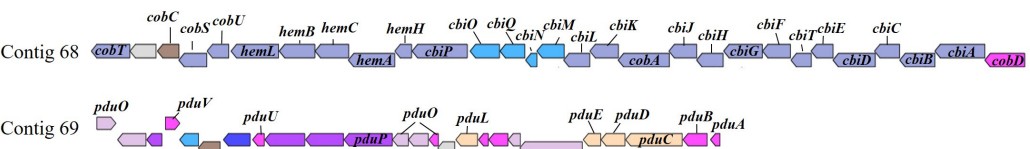

**Figure 6.** The schematic representation of the *pdu-cbi-cbo-hem* cluster of *L. reuteri* KGC1901. The genes encoding cobalamin synthetic genes were located in contig 68 (**upper**), and the genes encoding propanediol utilization genes were located in contig 69 (**lower**).

### 3.6. Antioxidant Activity

The DPPH radical scavenging activity of KGC1901 was 13%, higher than that of the reference strain, *L. rhamnosus* GG (9%) (Table 4). Moreover, in RAW 264.7 cells, LPS treatment stimulated NO production to 79 μM, whereas 1% KGC1901 CFS treatment suppressed NO production to 52.8 μM (Figure 7). This radical scavenging activity and NO

production inhibition activity may be useful features to reduce oxidative damage in the human body. Kim et al. studied the free-radical scavenging activity and NO production of 112 probiotic strains and 4 strains were selected [28]. NO production is part of the immune response stimulated by cytokines and microbial substances such as LPS and is regulated by the iNOS gene. NO plays immunologically and physiologically important roles; however, it can also be a reactive radical that causes inflammatory reactions. Therefore, the ability of KGC1901 to suppress NO production in LPS-stimulated RAW 264.7 cells may be useful for suppressing inflammation and oxidative damage. However, additional studies on related mechanisms such as the expression of inflammatory cytokines are needed to demonstrate anti-inflammatory functionality.

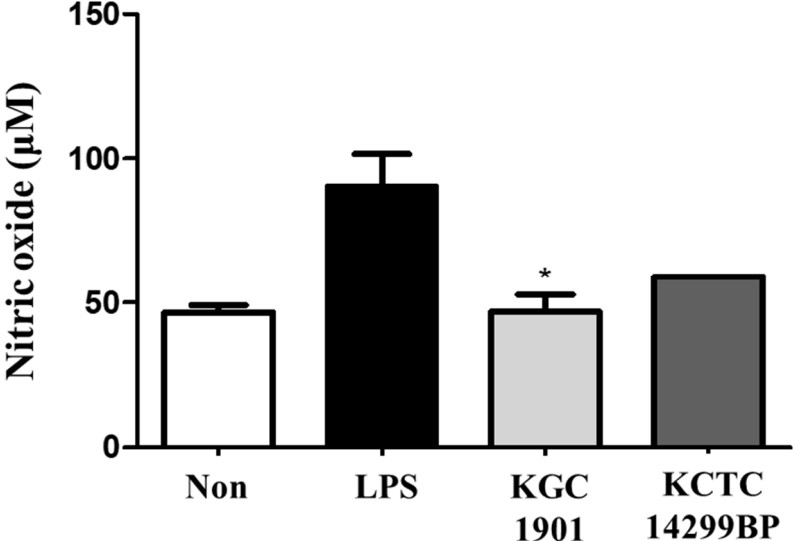

**Figure 7.** Nitric oxide production of *L. reuteri* KGC1901 in lipopolysaccharide-induced RAW 264.7 cells. Significant correlation between LPS and *L. reuteri* KGC1901 treatment is shown by asterisk (* $p < 0.05$).

## 4. Conclusions

The aim of this study was to isolate endophytic lactic acid bacteria isolated from the root of ginseng and to investigate the probiotic properties and safety of the isolate based on genomic and phenotypic characteristics as a potential probiotic strain. We also investigated antimicrobial activity against *C. difficile* and antioxidant activities. On the basis of our findings, it was determined that *Limosilactobacillus reuteri* KGC1901 isolated from ginseng is a potential probiotic to prevent disease related to *C. difficile* and oxidative stress.

**Author Contributions:** Conceptualization, S.-H.L. and S.-K.K.; formal analysis, S.-K.K.; investigation, H.-Y.Y., M.K. and Y.-S.L.; methodology, H.-Y.Y., M.K. and Y.-S.L.; project administration, S.-H.L. and S.-K.K.; supervision, S.-H.L.; writing—original draft preparation, H.-Y.Y.; writing—review and editing, S.-K.K. All authors have read and agreed to the published version of the manuscript.

**Funding:** This research received no external funding.

**Institutional Review Board Statement:** Not applicable.

**Informed Consent Statement:** Not applicable.

**Data Availability Statement:** The data presented in this study are available in the article.

**Conflicts of Interest:** The authors declare no conflict of interest.

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
