# Peer review of "Characterization of Limosilactobacillus reuteri KGC1901 Newly Isolated from Panax ginseng Root as a Probiotic and Its Safety Assessment"

_fermentation, doi:10.3390/fermentation9030228_

Round 1
Reviewer 1 Report
The interesting part of this study is the isolation of an endophytic probiotic lactic acid bacteria (Limosilactobacillus reuteri) from ginseng root. Authors have done some tests to estimate the safety risk and probiotic properties of the L. reuteri, including whole-genome analysis, antibiotic analysis, binding ability analysis, and other essential biochemical analysis, and the results preliminarily shown the bacteria is prebiotic and safe. I believe the authors want to find an autochthonous probiotic strain that assist in fermentation of ginseng to improve its function, therefore the investigation of the properties of safety and fermentation are necessary; in the study, the authors studied the safety and probiotic property of L. reuteri firstly. Overall, I think the paper is suitable for publication, however there are some aspects that, in my opinion, can be improved.
1. Check line 68, what dose “that f” mean?
2. In the part of “antimicrobial activity of KGC1901 CFS against CD (line 183 and 324)”, although the results demonstrated that the KGC1901 CFS can inhibit growth of CD, why the inoculation proportions of KGC1901 CFS are 10% and 20%? Is there any reference support or other considerations?
3. In the line 242-244, the authors have described the microbial enzymes may generate carcinogens or increase the possibility of carcinogenesis in the human body and the β-glucosidase and β-glucuronidase were proposed; however, the KGC1901 CFS were demonstrated have ability of producing the enzymes, and will the results increase the safety risk concerns of the strain? In my suggest, the functional and probiotic properties of these enzymes may be discussed.
4. Check line 378. The significant difference value p should be < 0.05 rather than < 0.5.
5. It’s better to perform the difference significance analysis and annotate the results in the data table (if applicable), such as Table 4.
6. The font of Latin name in phylogenetic tree (Fig.1) should be modified to italic.
Author Response
We are thankful to Reviewer 1 for his/her detailed and constructive comments and suggestions on our manuscript. We trust that these suggestions help us improve the quality of the manuscript. The appropriate changes made in the revised manuscript are highlighted. Our specific responses are following.
Comment #1:
Check line 68, what dose “that f” mean?
Response:
We really appreciate the reviewer’s comment. According to the reviewer’s comment, we checked the sentence. Those words, ‘that f’, are unnecessary so we have been deleted in the revised manuscript. (line 69 in the revised manuscript)
Comment #2:
In the part of “antimicrobial activity of KGC1901 CFS against CD (line 183 and 324)”, although the results demonstrated that the KGC1901 CFS can inhibit growth of CD, why the inoculation proportions of KGC1901 CFS are 10% and 20%? Is there any reference support or other considerations?
Response:
In previous study, Yang et al. studied the antimicrobial activities of L. reuteri AN417 cell-free supernatant (CFS) against oral pathogenic bacteria (Sci Rep 2021, 11, 1631). In this study, the minimal inhibitory volume of CFS was 10% to 30%. We actually tested the antimicrobial activity of 10%, 20%, and 40% of KGC1901 CFS against CD by referring to this previous study. Accordingly, based on the results of adding antimicrobial activity when 40% of KGC1901 CFS was added, Figure 5 was modified and the method was also modified. (line 191, and 359 in the revised manuscript)
Comment #3:
In the line 242-244, the authors have described the microbial enzymes may generate carcinogens or increase the possibility of carcinogenesis in the human body and the β-glucosidase and β-glucuronidase were proposed; however, the KGC1901 CFS were demonstrated have ability of producing the enzymes, and will the results increase the safety risk concerns of the strain? In my suggest, the functional and probiotic properties of these enzymes may be discussed.
Response:
As we mentioned in the original article, the enzymatic activities such as β-glucosidase and β-glucuronidase are important factors to be considered in terms of safety as probiotics. However, L. reuteri KGC1901 did not show such enzyme activities (Table 3).
Comment #4:
Check line 378. The significant difference value p should be < 0.05 rather than < 0.5.
Response:
We really appreciate the reviewer’s comment. According to the reviewer’s comment, ‘p < 0.5’ were changed to ‘p < 0.05’. (line 382 in the revised manuscript)
Comment #5:
It’s better to perform the difference significance analysis and annotate the results in the data table (if applicable), such as Table 4.
Response:
We really appreciate the reviewer’s comment. According to the reviewer’s comment, we performed the significance analysis between L. reuteri KGC1901 vs. LGG, and p-values were added in paragraph 3.3.
Comment #6:
The font of Latin name in phylogenetic tree (Fig.1) should be modified to italic.
Response:
We appreciate the reviewer’s comment. According to the reviewer’s comment, the font of Latin name in phylogenetic tree was changed to italic.

Reviewer 2 Report
The authors isolated a strain of Lactobacillus rhamnosus from Panax ginseng root and evaluated its safety and probiotic properties.
But there are problems with the manuscript as follows:
Line 55- The whole paper is about bacteria, so it doesn't make sense to describe Korean ginseng.
Line 212- Please add references
In paragraph 3.1, how many LABs were isolated from the root system of ginseng, and why L. reuteri KGC1901 was selected for research, please explain.
Line 239- What is KCTC3594, why did compare carbohydrate fermentation capacity with KCTC3594 ?
Table 3- According to what criteria, the enzyme activity is divided into no activity, poor activity, moderate activity, strong activity.
Line 291- What is ResFinder and VFDB ?

Author Response
We are thankful to Reviewer 2 for his/her positive comments on the manuscript. His/her comments have made our manuscript better representative of concepts and conclusions and greatly improved. The appropriate changes made in the revised manuscript are highlighted. Our specific responses are following.
Comment #1:
Line 55- The whole paper is about bacteria, so it doesn't make sense to describe Korean ginseng.
Response:
As you mentioned, this article is about bacteria, a Limosilactobacillus reuteri. Recently, various plant-derived functional lactic acid bacteria (LAB) have been studied worldwide. We used Korean ginseng (Panax ginseng) as a source to isolate useful LAB and described its general characteristics.
Comment #2:
Line 212- Please add references.
Response:
We appreciate the reviewer’s comment. According to the reviewer’s comment, we added reference. (line 217 in the revised manuscript)
Comment #3:
In paragraph 3.1, how many LABs were isolated from the root system of ginseng, and why L. reuteri KGC1901 was selected for research, please explain.
Response:
Our research team isolated six species of LAB from raw ginseng including Limosilactobacillus reuteri. In previous articles, we showed probiotic properties and health-beneficial effects of Lacticaseibacillus casei KGC1201 [20] (added in the revised manuscript) and Limosilactobacillus fermentum KGC1601 [6] isolated from Panax ginseng. Additionally, Supplementary Table S1 is added to explain reviewer 2’s request.
When it comes to L. reuteri, this species has various health beneficial functions such as antimicrobial activities, immunomodulatory effects, production of vitamins, lowering serum cholesterol level, etc. [13]. And the reason we selected L. reuteri KGC1901 is because it is the most suitable strain for commercial use in terms of safety and functionality.
Comment #4:
Line 239- What is KCTC3594, why did compare carbohydrate fermentation capacity with KCTC3594?
Response:
Limosilactobacillus reuteri KCTC3594 is a type strain purchased from Korea Collection for Type Culture (KCTC, Jungeup, Korea). We wanted to show the differences between KGC1901 and the type strain in the basic characteristics of the L. reuteri strains. We added the information of KCTC3594 in paragraph 2.2. (line 117-118 in the revised manuscript)
Comment #5:
Table 3- According to what criteria, the enzyme activity is divided into no activity, poor activity, moderate activity, strong activity.
Response:
The enzyme activity was divided according to the color reaction chart provided by API ZYM kit manufacturer. Each degree of activity was marked according to the changed color intensity as shown in Supplementary Figure S1. In addition, there were minor errors in the interpretation of the results, which were corrected. (Table 3 and line 252 in the revised manuscript)
Comment #6:
Line 291- What is ResFinder and VFDB?
Response:
‘Resfinder’ is an online bioinformatic tool for identification of antimicrobial resistance genes in next-generation sequencing data and prediction of phenotypes from genotypes. And the Virulence Factor Database (VFDB) is also an online resource for curating information about virulence factors of bacterial pathogens. Through these online databases, it was confirmed that the genome sequence of KGC1901 does not contain antibiotic resistance genes and virulence factors.
